# Systemic Lactate Elevation Induced by Tobacco Smoking during Rest and Exercise Is Not Associated with Nicotine

**DOI:** 10.3390/ijerph19052902

**Published:** 2022-03-02

**Authors:** Sri Sumartiningsih, Setya Rahayu, Eko Handoyo, Jung-Charng Lin, Chin Leong Lim, Michal Starczewski, Philip X. Fuchs, Chia-Hua Kuo

**Affiliations:** 1Department of Sports Science, Universitas Negeri Semarang, Gedung F1 Kampus Sekaran-Gunungpati, Semarang 50229, Indonesia; setyarahayu@mail.unnes.ac.id; 2Graduate School of Physical Education, Postgraduate Universitas Negeri Semarang, Gedung A Kampus Pascasarjana Jl. Kelud Utara III, Semarang 50237, Indonesia; eko.handoyo@mail.unnes.ac.id; 3Department of Political and Citizenship, Universitas Negeri Semarang, Gedung C Kampus Sekaran-Gunungpati, Semarang 50229, Indonesia; 4Department of Physical Education and Sport Sciences, National Taiwan Normal University, Taipei 111, Taiwan; normalin@ms34.hinet.net; 5Lee Kong Chian School of Medicine, Nanyang Technological University, Singapore 639798, Singapore; fabianlim@ntu.edu.sg; 6Faculty of Rehabilitation, Józef Piłsudski University of Physical Education, 00-809 Warsaw, Poland; michal.starczewski@awf.edu.pl; 7Department of Athletic Performance, National Taiwan Normal University, Taipei 116, Taiwan; philip.fuchs@ntnu.edu.tw; 8Laboratory of Exercise Biochemistry, College of Kinesiology, University of Taipei, Taipei 111, Taiwan

**Keywords:** e-cigarette, nicotine, vaping, tobacco, heart rate, glucose

## Abstract

Lactate is a metabolite produced during anaerobic glycolysis for ATP resynthesis, which accumulates during hypoxia and muscle contraction. Tobacco smoking significantly increases blood lactate. Here we conducted a counter-balanced crossover study to examine whether this effect is associated with inhaling nicotine or burned carbon particles. Fifteen male smokers (aged 23 to 26 years) were randomized into 3 inhalation conditions: tobacco smoking, nicotine vaping, and nicotine-free vaping, conducted two days apart. An electronic thermal evaporator (e-cigarette) was used for vaping. We have observed an increased blood lactate (+62%, main effect: *p* < 0.01) and a decreased blood glucose (−12%, main effect: *p* < 0.05) during thermal air inhalations regardless of the content delivered. Exercise-induced lactate accumulation and shuttle run performance were similar for the 3 inhalation conditions. Tobacco smoking slightly increased the resting heart rate above the two vaping conditions (*p* < 0.05), implicating the role of burned carbon particles on sympathetic stimulation, independent of nicotine and thermal air. The exercise response in the heart rate was similar for the 3 conditions. The results of the study suggest that acute hypoxia was induced by breathing thermal air. This may explain the reciprocal increases in lactate and decreases in glucose. The impaired lung function in oxygen delivery of tobacco smoking is unrelated to nicotine.

## 1. Introduction

Tobacco smoking is a risk factor for all-cause mortality and impaired pulmonary function [1,2,3,4]. Nicotine is widely regarded as the main addictive component of tobacco during smoking [5]. Breathing smoke from nicotine-containing tobacco cigarettes (1–14 mg) [6] or electronic (E) cigarettes (3–19 mg/mL) increased heart rate and blood pressure [7,8,9]. In addition to nicotine, a variety of carbon particles is generated during the burning process of tobacco leaves. Burned carbon particles and various chemicals in combination of thermal stress can also cause edema and injury to the lower airway and alveoli [10]. It is unclear whether nicotine stimulation or burned carbon particles are responsible for compromised oxygen delivery in the respiration system.

Lactate is a glycolic metabolite produced at hypoxia when oxygen is insufficiently supplied for aerobic ATP production. An acute episode of tobacco smoking increases the rate of lactate appearance in the blood [11]. These results implicate a potential hypoxia effect associated with a metabolic shift from aerobic to anaerobic substrates.

In this study, we hypothesized that an increased lactate accumulation in the blood after tobacco smoking at rest and following exercise is associated with the heated air from tobacco leaves, independent of nicotine. To address this question, a nicotine-containing aerosol and nicotine-free aerosol were also inhaled into the lungs of smokers via an e-cigarette thermal evaporator. The rate of glucose appearance increases during exercise [12]. Thus, blood levels of lactate and glucose were also assessed following tobacco smoking, nicotine vaping, and nicotine-free vaping conditions at rest and post-exercise compared with a non-smoking baseline.

## 2. Materials and Methods

### 2.1. Participants

A total of 20 eligible participants initially joined this study with five dropouts due to time constraints to comply with the testing schedule. Fifteen male smokers (averaged 9 cigarettes per day, 3.5 years, estimated 19 mg of nicotine per day) aged 23.9 ± 1.1 years (height 1.7 ± 0.1 m and weight 65 ± 8.8 kg) voluntarily participated in this study. All participants completed a medical history and an informed consent form before testing. The Faculty of Medicine Diponegoro University, Semarang, Indonesia, approved the study (Project no. 580/EC/FK-RSDK/IX/2017). All experimental procedures were conducted in accordance with the Declaration of Helsinki. Participants were given a full explanation of the purpose, testing procedure, and the potential risks of participation. All participants presented as free from cardiovascular diseases and diabetes mellitus according to their routine physical examination.

### 2.2. Study Design

This study was conducted using a randomized counter-balanced crossover design of three smoking conditions: nicotine free vaping (e-cigarette without nicotine), nicotine vaping (e-cigarette smoking with nicotine, 3 mg), and tobacco cigarette smoking (nicotine, 3 mg). 

### 2.3. Testing Protocol

All participants arrived at the laboratory in the morning from 09:00 a.m. to 12:00 p.m. after a 12-h fast (free of smoking and alcohol). After arrival, participants rested for 10 min. Lactate, glucose, and heart rate responses were then assessed at baseline (designated as Pre) under a sedentary condition. For each smoking trial, inhalation of the heated smoke lasted for 45 min. Blood samples were then collected immediately following the smoking session (designated as Post). Participants were blinded to the nicotine ingredients during the electronic cigarette smoking. Lactate, glucose, and heart rate responses to exercise (shuttle runs) were assessed 10 min after the smoking session. Each participant conducted three sessions of a Maximal Multistage 20 m Shuttle Run Test (MMST) at their best effort. Blood samples were collected again for assessment of exercise response (designated as Post).

### 2.4. MMST (Maximal Multistage 20 m Shuttle Run Test)

The MMST was designed to evaluate the exercise performance. The participants were required to run back and forth on a 20 m course and touch the 20 m line. A sound signal was emitted from a prerecorded tape at the same time. For the MMST, the participants were required to run until exhaustion, and the levels and shuttles were then calculated [13].

### 2.5. Lactate, Glucose, and Heart Rate

Lactate and glucose concentrations were assessed immediately following blood collection using Accutrend^®^ Plus (Roche Diagnostics, Rotkreuz, Switzerland). Heart rate (HR) was measured by a Polar RS800X Heart Rate Monitor Polar Electro (Kempele, Finland).

### 2.6. Statistics

All results were presented as mean ± standard error (SE). Type 1 error equal or less than 5% for comparing mean difference was considered significant. Two-way ANOVA with repeated measures was used to determine the main effect and interactive effects of intervention. The percentage (%) change after smoking and after exercise from baseline for lactate, glucose and HR were analyzed using paired t-test. Effect size was indicated by Cohen’s *d*, interpreted as small (*d* = 0.2), medium (*d* = 0.5), and large (*d* = 0.8) [14]. 

## 3. Results

### 3.1. Blood Lactate Response

Figure 1 shows the blood lactate response to three inhalation conditions at rest (Figure 1A) and exercise (Figure 1B). At rest, blood lactate increased after tobacco smoking, nicotine-free vaping, and nicotine vaping to a similar extent (*d* = 1.1, main effect of time: *p* < 0.001). 

Following a standard shuttle run protocol, the performance times for the three post-inhalation conditions were comparable (nicotine free vaping, 347 ± 33 s; nicotine vaping, 349 ± 33 s; tobacco smoking, 325 ± 30 s). Blood lactate increased above the non-smoking baseline (Pre). Lactate accumulation after exercise was similar for all three conditions (nicotine free vaping, 10.0 ± 0.6 mM; nicotine vaping, 11.0 ± 0.5 mM; tobacco smoking, 12.0 ± 0.4 mM) (*d* = 0.96, main effect of time, *p* < 0.001).

**Figure 1 ijerph-19-02902-f001:**
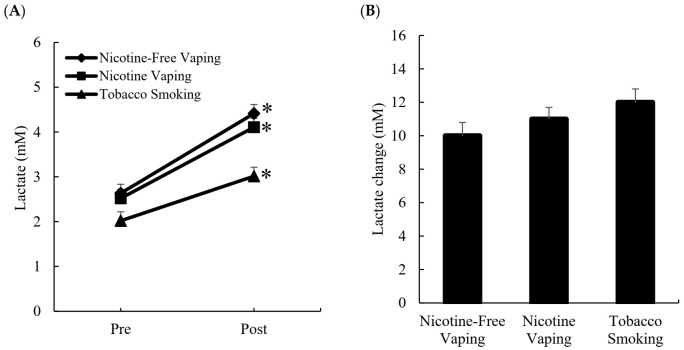
Blood lactate increases after smoking at rest and exercise. Blood lactate levels elevated for the three inhalation conditions were similar at rest (**A**). Exercise-induced increases in blood lactate levels from non-inhalation baseline were also similar among the three inhalation conditions (**B**). Intervention: Nicotine-free vaping (0 mg); nicotine vaping (nicotine: 3 mg); tobacco cigarette (nicotine: 3 mg). Main effect of intervention: *p* < 0.01 (two-way ANOVA). Main effect of exercise: *p* < 0.001 (two-way ANOVA). * Significant difference versus nicotine-free vaping condition.

### 3.2. Blood Glucose Response

Figure 2 shows blood glucose response to three inhalation conditions at rest (Figure 2A) and exercise (Figure 2B). At rest, blood glucose decreased after smoking, and was similar for the tobacco smoking (*d* = 0.8, *p* < 0.01), nicotine-free vaping (*d* = 1.6, *p* < 0.01), and nicotine vaping (*d* = 1.2, *p* < 0.01) conditions. 

Following a standard shuttle run protocol, blood glucose decreased below the non-inhalation baseline (Pre). This decrease was similar for the three inhalation conditions (nicotine free vaping, 25.3 ± 11.9 mg/dL, nicotine vaping, 27.4 ± 11.4 mg/dL, tobacco smoking, 29.0 ± 11.6 mg/dL, *d* = 0.9, main effect of time: *p* < 0.01).

**Figure 2 ijerph-19-02902-f002:**
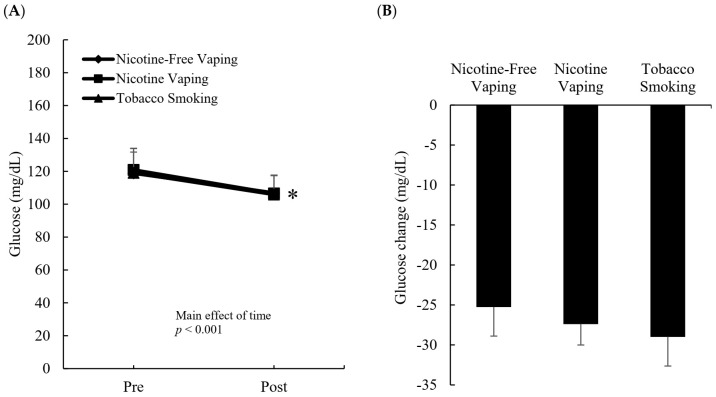
Blood glucose responses to smoking at rest and exercise. Blood glucose levels at rest decreased to a similar level for all inhalation conditions (**A**). Exercise-induced decreases in blood glucose from non-smoking baseline were similar after all inhalation conditions (**B**). Intervention: Nicotine-free vaping (0 mg); nicotine vaping (nicotine: 3 mg); tobacco cigarette (nicotine: 3 mg). Main effect of smoking: *p* < 0.01 (two-way ANOVA). Main effect of exercise: *p* < 0.001 (two-way ANOVA). * Significant difference versus nicotine-free vaping.

### 3.3. Heart Rate Response

Figure 3 shows the heart rate response to the three inhalation conditions at rest (Figure 3A) and exercise (Figure 3B). Resting heart rate significantly increased (+12%) after tobacco smoking (*d* = 0.8, *p* = 0.041). Both nicotine vaping and non-nicotine vaping had no significant effect (−3%) on resting heart rate.

Following the standard shuttle run protocol, heart rate increased significantly for the tobacco smoking (89 ± 2.8 beat/min), nicotine vaping (93 ± 2.8 beat/min), and non-nicotine (95 ± 2.8 beat/min) conditions. No difference in the post-exercise heart rate was found among the three conditions.

**Figure 3 ijerph-19-02902-f003:**
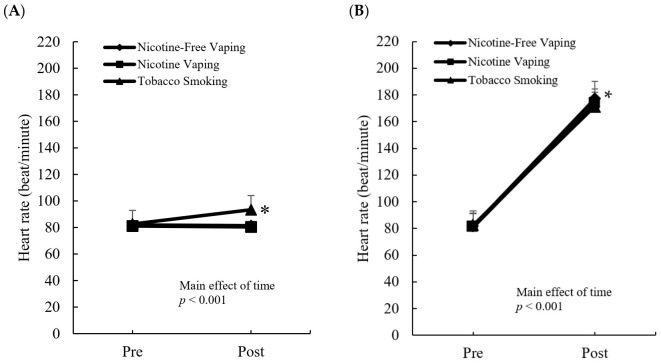
Heart rate responses to smoking at rest and exercise. Tobacco smoking significantly elevated resting heart rate in comparison with both nicotine and nicotine-free vaping conditions (**A**). Exercise-induced increases in heart rate from non-smoking baseline were similar for tobacco smoking, nicotine vaping, and nicotine-free vaping conditions (**B**). Intervention: Nicotine-free vaping (0 mg); nicotine vaping (nicotine: 3 mg); tobacco cigarette (nicotine: 3 mg). Main effect of exercise: *p* < 0.001 (two-way ANOVA). * Significant difference from the e-cigarette vaping (nicotine or nicotine-free) conditions, *p* < 0.05.

## 4. Discussion

The present study was designed to examine whether the tobacco smoking-induced blood lactate increases are mediated by nicotine [11,15], and whether this influenced blood glucose and exercise performance. In this study, we found an increased blood lactate accumulation together with decreased blood glucose after nicotine-free vaping (0 mg), nicotine vaping (nicotine: 3 mg), and tobacco cigarette smoking (nicotine: 3 mg). Furthermore, tobacco smoking significantly increased heart rate in comparison with e-cigarette smoking containing nicotine. Hypoxia is known to increase glucose transport in skeletal muscle [16] and cardiac muscle [17]. Therefore, the findings of the study suggest that (1) tobacco smoking induces an acute systemic hypoxia associated with impaired lung function in oxygen delivery, independent of nicotine; (2) autonomic modulation, reflected by an increased resting heart rate, is associated with the inhalation of burned carbon particles during tobacco smoking; (3) nicotine is not associated with the heart rate increase during tobacco smoking; (4) exercise performance seems to be unaffected by an acute session of tobacco smoking among the young smokers. 

As a consequence of tobacco smoking, carbon monoxide has been widely assumed to cause hypoxia. This assumption is based on increased carbon monoxide observed in tobacco cigarette smokers than in non-smokers [18,19,20,21]. This is also strengthened by the fact that chronic tobacco smokers show a greater carboxyhemoglobin concentration level than acute tobacco smokers [22]. However, in this study, the magnitude of the blood lactate increase was similar regardless of ingredients, suggesting that the thermal air inhalation is the main cause of impaired oxygen delivery in the lungs, not nicotine or burned carbon particles. Lactate is an anaerobic metabolite produced during glycolysis, which is a good metabolic marker of tissue hypoxia [23,24]. Hypoxia is known to increase glucose uptake by skeletal muscle [25]. Since the amount of carbon monoxide produced by nicotine-free e-cigarettes is neglectable [19], the hypoxia effect of tobacco smoking is unlikely to be caused by carbon monoxide alone. 

Burned carbon particles during tobacco smoking may contribute to the observed increased heart rate. Heart rate is directly associated with the magnitude of sympathetic activation or parasympathetic inhibition to alter oxygen delivery in circulation [26]. The finding of an increased heart rate on tobacco smoking indicates a modulation in the autonomic nervous system. Both e-cigarettes with and without nicotine have no effect on heart rate. Taken together, the influence of tobacco smoking on increased heart rate is likely associated with carbon particles from burned tobacco leaves, independent of thermal air and nicotine. This is supported by a previous report that the inhalation of fine carbon particles alters heart rate and heart rate variability in diabetes patients [27]. Furthermore, the present study also demonstrated that the increase in heart rate after smoking is not due to nicotine alone. In this study, we could not preclude the possibility that higher doses of nicotine may produce more pronounced effects in the heart rate response at rest and exercise [28]. Furthermore, an increase in heart rate after intravenous nicotine infusion was reported previously [29]. Therefore, the isolated effect of nicotine should be noted. The difference on heart rate between tobacco smoking and e-cigarette smoking (with and without nicotine) diminished after exercise at maximal effort, suggesting that exercise at such an intensity has to activate the sympathetic nervous system towards a maximal level. Despite a significant elevation in the resting heart rate after tobacco smoking, the shuttle run performance was not negatively influenced. This result is consistent with several previous reports in athletes [30,31]. 

The current data suggest that nicotine as an addictive component of cigarette is not the only determinant for changes in lactate, glucose, and heart rate. Other contributing factors are likely to be the burned carbon particles and thermal air inhalation. Nevertheless, adverse effects of nicotine itself must be recognized (e.g., on heart rate, blood pressure, and perceived smoking urge) [29]. The specific effects of nicotine itself as well as effects of smoking in general are highly important in the context of public health. Tobacco smoking is associated with increased levels of toxic metals [32] and chemicals [33], and the significance for public health and health care systems is well-known [34]. E-cigarettes also include numerous harmful components besides nicotine (e.g., formaldehyde, acetaldehyde, and acroleine) that we have not investigated in the study. E-smoking also has a harmful impact on human health (e.g., increase in impotence, peripheral airway flow resistance, and oxidative stress) [35] that is relevant for public health.

One limitation of the study is that the knowledge generated in the acute study cannot be generalized to chronic smoking or smoking higher nicotine doses. The harmful effect of chronic tobacco cigarette smoking on mortality has been well-documented in the past [36,37]. In addition, the study did not include a non-vaping control and, therefore, cannot preclude a possibility that thermal vaping (free of nicotine) may negatively impact exercise performance.

## 5. Conclusions

Inhalation of heated air into human lungs significantly increased blood lactate and decreased blood glucose, suggesting a transient hypoxia associated with a declined lung function for oxygen delivery. This metabolic response was not fully explained by nicotine and burned carbon particles alone. Nevertheless, adverse effects of nicotine on human health should not be underestimated. Moreover, smoking involves many other toxic components and is well-known for its significance for public health and health care policies.

## Data Availability

Available upon request.

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
