# Peer review of "Systemic Lactate Elevation Induced by Tobacco Smoking during Rest and Exercise Is Not Associated with Nicotine"

_ijerph, 2022, doi:10.3390/ijerph19052902_

Round 1

Reviewer 1 Report

I am grateful for the opportunity to review the interesting topic „Systemic lactate elevation induced by tobacco smoking during rest and exercise is not associated with nicotine“.

I read the author's article with pleasure and great interest. The manuscript is very interesting and well-prepared, even though it announces controversial results. However, I have minor observations regarding the further applicability of the results obtained.

As we know nicotine is a dangerous and highly addictive chemical. It can cause an increase in blood pressure, heart rate, flow of blood to the heart and a narrowing of the arteries. Nicotine may also contribute to the hardening of the arterial walls, which in turn, may lead to a heart attack.

Despite this, the study demonstrated that the effect of heart rate increase after smoking is unrelated with nicotine. There may be a need for a more detailed interpretation of the results obtained depending on the context of public health.

In summary, I would recommend that authors anticipate the significance and practical application of the findings identified. It might be helpful to link findings not with smoking but specifically with nicotine use, physical activity levels and health-friendly and health-unfriendly determinants. Additional discussion may potentially increase the completeness of the content of the article.

My observations are minor. It is recommended to accept the manuscript.

Best Regards

Author Response

Thanks for reviewer's time and effort. We agree with the reviewer that nicotine has clear adverse effects, and we confirm that the current findings of this manuscript deserve consideration in the context of public health. The second aspect of the reviewer’s comment refers to the significance and practical application of the findings, linking specifically with nicotine in the context of health and physical activity. As per reviewer’s suggestion, we strengthened the discussion (adaptation in the third paragraph and a new fourth paragraph) and conclusion sections by elaborating on both aspects outlined by the reviewer. We also added references to support the additions.

Reviewer 2 Report

Very interesting article regarding the pre- and post-workout levels of lactate and glucose for three types of smoking (vaping with nicotine, vaping without nicotine, and smoking cigarettes). Very minor suggestions below:

Abstract

Line 29: Consider "...regardless of the content delivered."

Lines 34-35: This sentence is worded strangely, consider revising for clarity

Line 36: Consider "...is unrelated to nicotine."

Introduction

Line 42: Do we need to include the word "leaves?" It might be implied simply by saying "...the main stimulant component of tobacco during smoking."

Line 45: Consider "...tobacco leaf."

Lines 47-48: This sentence is worded strangely, consider revising for clarity

Materials and Methods

Line 74: Consider "All participants presented as free from cardiovascular disease..."

Line 84: Unclear what is meant by "a 12-hour fast...a 10-min rest." 

Line 86: Consider "...inhalation of the heated smoke" (instead of "smokes")

Line 88: Consider "Participants were blind to..."

Results

Line 112-113: Consider "...and after exercise (B)."

Results section 3.2 is very well-worded and concise, consider revising results section 3.1 for parallel construction and to aid in clarity.

Figure 2

Consider revising the phrase "...elevated resting heart rate above both nicotine..." Maybe something like "...elevated resting heart rate for both nicotine..."

Discussion

Line 162: Consider "...whether this influenced..."

Line 165: Consider "...tobacco cigarette smoking (nicotine: 3mg)."

Line166: Same as above, consider revising "increased heart rate above" to something more colloquial, maybe "increased heart rate more than..." 

Line 174-176: These sentences are worded strangely, consider revising for clarity

Line 183: Consider "...free e-cigarettes is..."

Line 183: Consider "...smoking is unlikely to be..."

Lines 185-186: This sentence is worded strangely, consider revising for clarity

Line 196-197: Consider "...is unrelated to nicotine."

Line 197: Consider "...higher doses of nicotine..."

Line 198: Consider "...may produce more pronounced effects in heart rate...."

Line 206: Consider "...chronic smoking or smoking higher nicotine doses."

Line 208: Consider "...the study design did not have a non-vaping control..."

Author Response

Thank you for the very precise suggestions. We addressed all of them and implemented changes accordingly.

Abstract

Line 29: Consider "...regardless of the content delivered."
A: Corrected after suggestion. Highlighted in red.

Lines 34-35: This sentence is worded strangely, consider revising for clarity
A: Corrected after suggestion. Highlighted in red.

Line 36: Consider "...is unrelated to nicotine."
A: Corrected after suggestion. Highlighted in red.

Introduction

Line 42: Do we need to include the word "leaves?" It might be implied simply by saying "...the main stimulant component of tobacco during smoking."
A: Corrected after suggestion. Highlighted in red.

Line 45: Consider "...tobacco leaf."
A: Corrected after suggestion. Highlighted in red.

Lines 47-48: This sentence is worded strangely, consider revising for clarity.
A: Corrected after suggestion. Highlighted in red.

Materials and Methods

Line 74: Consider "All participants presented as free from cardiovascular disease..."
A: Corrected after suggestion. Highlighted in red.

Line 84: Unclear what is meant by "a 12-hour fast...a 10-min rest." 
A: Corrected after suggestion. Highlighted in red.

Line 86: Consider "...inhalation of the heated smoke" (instead of "smokes")
A: Corrected after suggestion. Highlighted in red.

Line 88: Consider "Participants were blind to..."
A: Corrected after suggestion. Highlighted in red.

Results

Line 112-113: Consider "...and after exercise (B)."
A: Corrected after suggestion as well as in the similar sentences in 3.2 and 3.3.

Results section 3.2 is very well-worded and concise, consider revising results section 3.1 for parallel construction and to aid in clarity.
A: Corrected after suggestion. Highlighted in red.

Figure 2

Consider revising the phrase "...elevated resting heart rate above both nicotine..." Maybe something like "...elevated resting heart rate for both nicotine..."
A: Corrected after accordingly. Highlighted in red.

Discussion

Line 162: Consider "...whether this influenced..."
A: Corrected after suggestion. Highlighted in red.

Line 165: Consider "...tobacco cigarette smoking (nicotine: 3mg)."
A: Corrected after suggestion. Highlighted in red.

Line166: Same as above, consider revising "increased heart rate above" to something more colloquial, maybe "increased heart rate more than..." 
A: Corrected after suggestion. Highlighted in red.

Line 174-176: These sentences are worded strangely, consider revising for clarity.
A: Reworded after suggestion. Highlighted in red.

Line 183: Consider "...free e-cigarettes is..."
A: Corrected after suggestion. Highlighted in red.

Line 183: Consider "...smoking is unlikely to be..."
A: Corrected after suggestion. Highlighted in red. In addition, we changed “contributed by” to “caused by … alone” to be correct because, strictly speaking, the finding does not allow to assess the contribution.

Lines 185-186: This sentence is worded strangely, consider revising for clarity
A: Revised after suggestion. Highlighted in red.

Line 196-197: Consider "...is unrelated to nicotine."
A: Corrected after suggestion. Highlighted in red.

Line 197: Consider "...higher doses of nicotine..."
A: Corrected after suggestion. Highlighted in red.

Line 198: Consider "...may produce more pronounced effects in heart rate...."
A: Corrected after suggestion. Highlighted in red.

Line 206: Consider "...chronic smoking or smoking higher nicotine doses."
A: Corrected after suggestion. Highlighted in red.

Line 208: Consider "...the study design did not have a non-vaping control..."
A: Corrected after suggestion. Highlighted in red.